# Appropriate data segmentation improves speech encoding models: Analysis and simulation of electrophysiological recordings

Ole Bialas[1,2]*, Edmund C. Lalor[1,2,3]*

**1** Department of Biomedical Engineering, University of Rochester, Rochester, New York, United States of America, **2** Del Monte Institute for Neuroscience , University of Rochester, Rochester, New York, United States of America, **3** Center for Visual Science, University of Rochester, Rochester, New York, United States of America

* obialas@ur.rochester.edu (OB); elalor@ur.rochester.edu (CL)

**Data availability statement:** All files are available from the OpenNeuro database (doi:10.18112/openneuro.ds004408.v1.0.8).

## Abstract

**Background**: In recent decades, studies modeling the neural processing of continuous, naturalistic, speech provided new insights into how speech and language are represented in the brain. However, the linear encoder models commonly used in such studies assume that the underlying data are stationary, varying to a fixed degree around a constant mean. Long, continuous, neural recordings may violate this assumption leading to impaired model performance. We aimed to examine the effect of non-stationary trends in continuous neural recordings on the performance of linear speech encoding models.

**Methods**: We used temporal response functions (TRFs) to predict continuous neural responses to speech while splitting the data into segments of varying length, prior to model fitting. Our Hypothesis was that if the data were non-stationary, segmentation should improve model performance by making individual segments approximately stationary. We simulated and predicted stationary and non-stationary recordings to test our hypothesis under a known ground truth and predicted the brain activity of participants who listened to a narrated story, to test our hypothesis on actual neural recordings.

**Results**: Simulations showed that, for stationary data, increasing segmentation steadily decreased model performance. For non-stationary data however, segmentation initially improved model performance. Modeling of neural recordings yielded similar results: segments of intermediate length (5–15 s) led to improved model performance compared to very short (1–2 s) and very long (30–120 s) segments.

**Conclusions**: We showed that data segmentation improves the performance of encoding models for both simulated and real neural data and that this can be explained by the fact that shorter segments approximate stationarity more closely. Thus, the common practice of applying encoding models to long continuous segments of data is suboptimal and recordings should be segmented prior to modeling.

**Funding:** The author(s) received no specific funding for this work.

**Competing interests:** The authors have declared that no competing interests exist.

## Introduction

Traditionally, studies on the neural processing of speech and language relied on repetitive presentations of prototypical words or sentences that are carefully manipulated along a single dimension of interest [1]. However, while those studies were successful in probing and identifying different modules of speech processing, their artificial and univariate approach makes them incapable of capturing the multi-stream, distributed nature of speech processing [2]. One way to overcome this limitation is to record brain responses to continuous narrative stories that present speech in its natural complexity. However, analyzing these recordings requires statistical models that can dissect activity arising from the different concurrent processes that contribute to the recorded activity.

In recent decades studies on the neural processing of speech increasingly used encoding models that predict the ongoing brain activity from (a combination of) acoustic, phonetic and semantic speech features [1,2]. Here, we focus on temporal response functions (TRFs) because they are comparatively small models can be easily interpreted and because they are used widely within the speech community.

The TRF approach treats the brain as a linear, time-invariant (LTI system) and tires to estimate the response characteristics of this system via deconvolution [3]. While, in theory, the TRF method works on indefinitely long continuous data, in practice, the data is usually divided into multiple segments. However, there exists no consensus or standard procedure with respect to the duration of those segments. While many studies analyze responses to segments of narrative stories on the order of minutes [4–6], others use shorter segments lasting tens of seconds [7] or even single sentences [8]. The way the data is segmented usually depends on the experimental process (e.g., whether the participants listen to individual sentences or long segments of a story) and how this affects the model's outcome is not considered. This is somewhat negligent because there are several ways in which data segmentation may affect the performance of TRF models.

### Effects of data segmentation

TRFs are linear models that map specific stimulus features to the ongoing brain response. This mapping is estimated by multiplying the covariance matrix for predictor (e.g. speech envelope) and estimand (e.g. neural recording) with the predictor's autocovariance matrix (see Eq 2). When the TRF is estimated across multiple segments, the covariance matrices are averaged across all segments and the TRF is obtained from the average matrices. The validity of this procedure hinges on the assumption that this converges on a stable estimate of the average covariance matrix. However, if there are only few segments, a single outlier may substantially alter the average. If the number of segments is larger, the effect of any single one is reduced. Thus, segmentation regulates overfitting to extreme values.

Segmentation also affects model fitting. Typically, fitting a TRF involves optimizing the regularization parameter $\lambda$ which penalizes large model weights and prevents overfitting. The optimal value for $\lambda$ is found by testing multiple candidate values by randomly splitting the data into train and test set, fitting the TRF on the former and evaluating its accuracy by predicting the latter. This requires that both test and train set are representative of the overall trends in the data so that the model can generalize from one set to the other. Representative subsets may be obtained more reliably if many short segments are randomly sampled from the recording.

Of course, segmenting the data into ever shorter bits may impair model accuracy if, at some point, the segments become too short to reliably estimate the covariance matrices. With too little data, the individual covariance matrices might represent the noise in the respective

segment, rather than the overall trend in the data. Finally, a key reason for why segment duration matters is that the TRF method treats the brain as a linear time-invariant (LTI) system which implicitly assumes that the modeled signal is a stationary process whose statistical properties are stable across time such that the particular time we observe it is of no relevance [9].

## Are neural recordings stationary?

Several studies examined the stationarity of EEG recordings by testing whether their amplitude distribution is Gaussian [10,11] or whether observed sequences of values above or below the mean are longer than would be expected by chance [12,13]. They agreed that, above a certain duration, EEG recordings could not be considered stationary, even though the critical duration varied between 2 and 25 seconds (for a review, see [14]). One study found no difference in the (non-)stationarity between EEG recordings during auditory stimulation, a cognitive task, and resting state, suggesting that these properties are mainly determined by the spontaneous portion of the EEG signal [15]. However, in lack of a ground truth, the origin of non-stationary trends is difficult to assert, and it is of minor importance to the question of appropriate data segmentation.

One can also reason about the stationarity of neural recordings based on the signals' power spectra. It is generally acknowledged that neural recordings exhibit a 1/f-like power spectrum, meaning that there is an inverse linear relationship between log power and log frequency [16–18]. This leads to the contradiction that a 1/f-process can not be stationary, since Rayleigh's energy theorem states that a signal's energy is equal to the integral of the power spectrum from negative to positive infinity. For a 1/f-distribution, however, the integral diverges because power approaches infinity as the frequency approaches 0. The alternative is to consider the signal as a non-stationary process where the signal can be integrated over a discrete time and frequency range but a single time-independent energy value can not be obtained [19,20].

In summary, neural time-series exhibit temporal and spectral characteristics of non-stationary processes. This is intuitively plausible, since the brain is a dynamic system where complex networks interact across multiple temporal and spatial scales and the signals recorded from this system mix with noise from various physiological and artificial sources. Our approach differs from the above cited studies in that we do not test non-stationarity directly but instead estimate the effect of data segmentation on the accuracy of models that assume stationarity. By combining this empirical test with generative simulations, we can show that the observed effect of segmentation can be explained by non-stationary trends in the data.

## Optimal segmentation

Taken together, the above points suggest that there is an optimal segment duration, where the data can be considered approximately stationary and the effect of outlier segments is suppressed while the individual segments are still long enough to reliably estimate their covariance and autocovariance matrices. Here, we systematically investigate the effect of data segmentation on the performance of TRF models. We use generative simulations to demonstrate that data segmentation improves prediction accuracy in the presence of non-stationary noise. Furthermore, we analyze EEG recordings of neural responses to continuous naturalistic speech and show that segmentation substantially improves prediction accuracy for many participants without negatively affecting the others. We thus recommend that future studies that use TRF models to predict EEG responses to continuous speech use segments of about

10 seconds. We don't necessarily mean to suggest that the experiment needs to be organized in 10 second trials, rather that the data should be restructured prior to model fitting.

## Materials and methods

Our central hypothesis is that dividing continuous neural recordings into shorter segments should improve model accuracy if the data are non-stationary. To test this hypothesis we fit our models on the same data while dividing them into segments of varying length. We use a generative simulation to test our hypothesis under a known ground truth and analyze EEG responses to continuous speech to test our hypothesis on actual neural data.

### 0.1 Data and code availability

The raw data are hosted on OpenNeuro and can be obtained, together the code for all simulations and analyses, from the Github repository for this study.

### 0.2 Ethics

This work used publicly available data. The original recordings were acquired in two unrelated studies [4,5] at Trinity College Dublin where they were approved by the Ethics Committee of the School of Psychology. No additional data were recorded for this study.

### Temporal response function

Under our model, the observed neural response is expressed as the weighted sum of the stimulus feature across multiple time lags, defined by the equation:

$$r(t) = \sum_n \sum_\tau w(\tau, n)\, s(t - \tau, n) + \epsilon(t) \tag{1}$$

Where $r(t)$ is the response observed at time $t$ and $s(t - \tau, n)$ is the stimulus feature at time lag $\tau$. The effect of the feature at time lag $\tau$ is given by the weight $w(n, \tau)$ and the error term $\epsilon(t)$ contains the residual response unexplained by the model. The vector $w$ that contains the weights across all time lags is called the temporal response function (TRF). The TRF is obtained via regularized regression, which, in matrix notation, can be written as:

$$w = (S^T S + \lambda I)^{-1} S^T r \tag{2}$$

Where $(S^T S)^{-1}$ is the inverted autocovariance matrix of the stimulus, $S^T r$ is the covariance matrix of stimulus and response and $\lambda I$ is a diagonal regularization matrix. The value of $\lambda$ is optimized for model accuracy, defined as the average correlation (Pearson's r) between the observed response $r(t)$ and its prediction $\hat{r}(t)$ which is obtained by convolving stimulus features and TRF:

$$\hat{r}(t) = \sum_n \sum_\tau s(t - \tau, f)\, w(\tau, n) \tag{3}$$

A more detailed mathematical description of the TRF, including multi-variable models, can be found in [3].

## Modeling EEG responses to speech

We analyzed data from 19 healthy, neurotypical adults (age 19-38, 13 male) who listened to a reading of roughly the first hour of "The Old Man and the Sea" by a single American male speaker. The prose in this part of the story is mostly descriptive and focuses on the protagonists environment, actions, and thoughts. It's narrated in the third person and the narrators tone alternates between somber reflection and subtle optimism. The story was divided into 20 segments, each lasting between 170 and 200 seconds. As the participants listened, their brain activity was recorded using a 128-channel ActiveTwo EEG system (BioSemi) at a sampling rate of 512 Hz. The data , which were collected in unrelated studies [4,5], are publicly available and no additional data was collected for the purpose of this study. We chose this data set because it has proven to be well suited for TRF analyses and the individual recordings are long enough to test our models across a wide range of segment durations.

We cropped all 20 segments to a duration of 120 seconds, resulting in 50 minutes of data per subject We filtered the segments between 1 and 20 Hz using a non-causal hamming-window band pass filter and downsampled to 64 Hz using MNE-Python [21]. Next, we used the random sample consensus algorithm to identify channels as bad if they were poorly predicted by their neighbors [22], interpolated them using spherical splines and re-referenced all channels to the global average.

We predicted the EEG responses from the spectro-temporal characteristics of the stimulus by band-pass filtering the audio material between 20 and 9000 Hz, dividing it into log-spaced bands [23] and computing the absolute Hilbert envelope for each spectral band, all using the soundlab Python package [24]. To test how the number of parameters and the degree of spectral detail affect model accuracy, we represented the stimulus with 1, 8 and 16 spectral bands and repeated the modeling procedure for each representation.

The TRF is fit to the data by testing multiple candidate values for $\lambda$ and selecting the one that maximizes prediction accuracy. For each of nine log-spaced values between $10^{-5}$ and $10^3$, the data segments are repeatedly split into train and test set using 5-fold cross validation. For each split, the TRF is estimated on the train set and evaluated by predicting the test set. Model accuracy for a given value of $\lambda$ is obtained by averaging prediction accuracy across all channels and splits. All modeling was done using the mTRFpy toolbox [25].

To estimate the effect of data segmentation on model accuracy, we repeat the above process while dividing the same 50 minutes of EEG recordings from every participant into segments of length 120, 60, 40, 30, 15, 10, 5, 2 and 1 second(s). The segment durations were chosen so that the 50 minutes of data could be evenly divided into segments for every duration. We excluded the data from further analysis if the accuracy of the best model was below the threshold of $r = 0.01$. We deliberately chose a low threshold to only reject participants where the model explained effectively none of the variance in the data. In our sample, this affected one participant for whom prediction accuracy approximated 0 across all models and segment durations (mean $r = 0.001$).

### 0.3 Simulation

We use a simple generative model to test the accuracy of TRF models under different conditions. We define the transfer function for simulating the neural response using a Gabor wavelet, obtained by modulating a sinusoidal with a Gaussian function:

$$w(t) = e^{-\frac{t-\mu^2}{2\sigma^2}} \cos 2\pi f t \tag{4}$$

Where $\mu$ and $\sigma$ are the mean and standard deviation of the Gaussian, of is the sinusoidal's frequency and $t$ is time. The blue curve in Fig 1a shows an example of such a wavelet with $\mu$ = 0.1 s, $\sigma$ = 0.01 s and $f$ = 4 Hz.

We simulate a stimulus by generating a sequence of randomly spaced square pulses, where the free parameters are the range of amplitude and width of the pulses (Fig 1b, blue line). Then, we convolve this stimulus with the wavelet kernel (Fig 1b, green line) and add noise (Fig 1c) to obtain the simulated neural response as defined in Eq 1. Finally, we fit a TRF to reconstruct the original wavelet (Fig 1a green line). To estimate the accuracy of the reconstructed TRF, we generate a new stimulus sequence and convolve it with the original and reconstructed TRF and calculate Pearson's correlation coefficient for the two responses (Fig 1d).

To determine how violating the assumption of stationarity affects model accuracy, we compare a scenario where the added noise is Gaussian to one where the noise has a 1/f distribution. 1/f noise is generated by computing the Fourier transform of white noise, imposing a 1/f distribution and computing the inverse Fourier transform of the modulated spectrum (for examples of Gaussian and 1/f noise see panels a and b of Fig 2). Note that the simulation represents the best-case scenario where non-stationarity is only introduced by the added background noise. If non-stationary background noise impairs model performance, this problem would certainly be amplified if the generating process itself would vary over time.

For both simulations, we divide the data into segments of 200, 100, 50, 25, 10, 5, 2 and 1 second(s), fit a TRF to 1000 seconds of training data and evaluate it on another 1000 seconds of testing data generated from the same model. Since the model is a stochastic process, we repeated the simulation 1000 times.

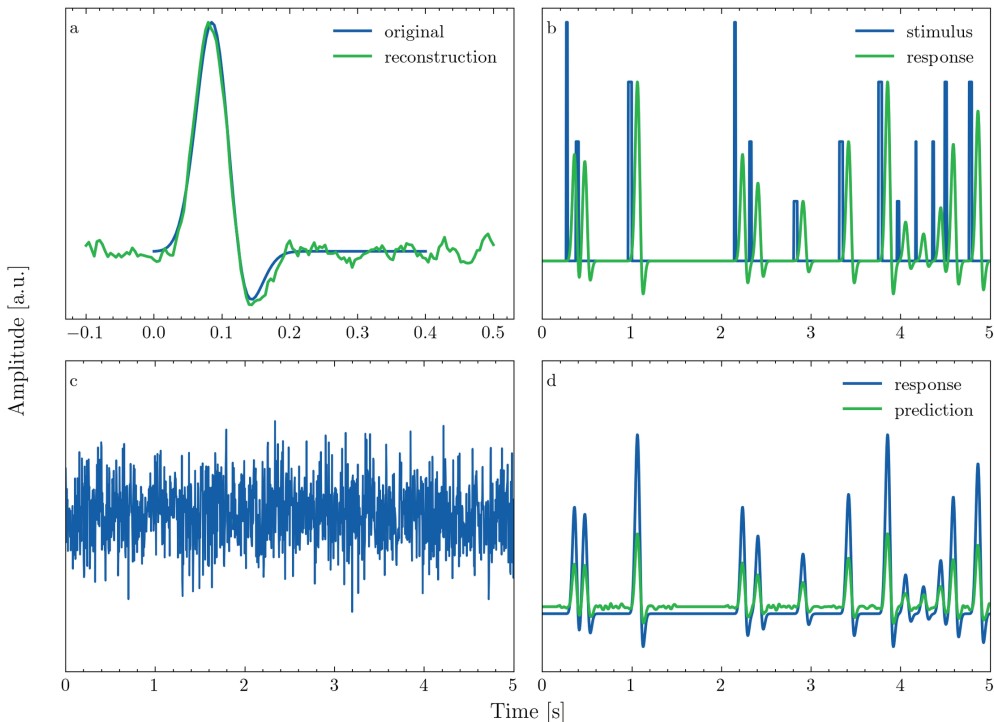

**Fig 1. Generative simulation framework.**

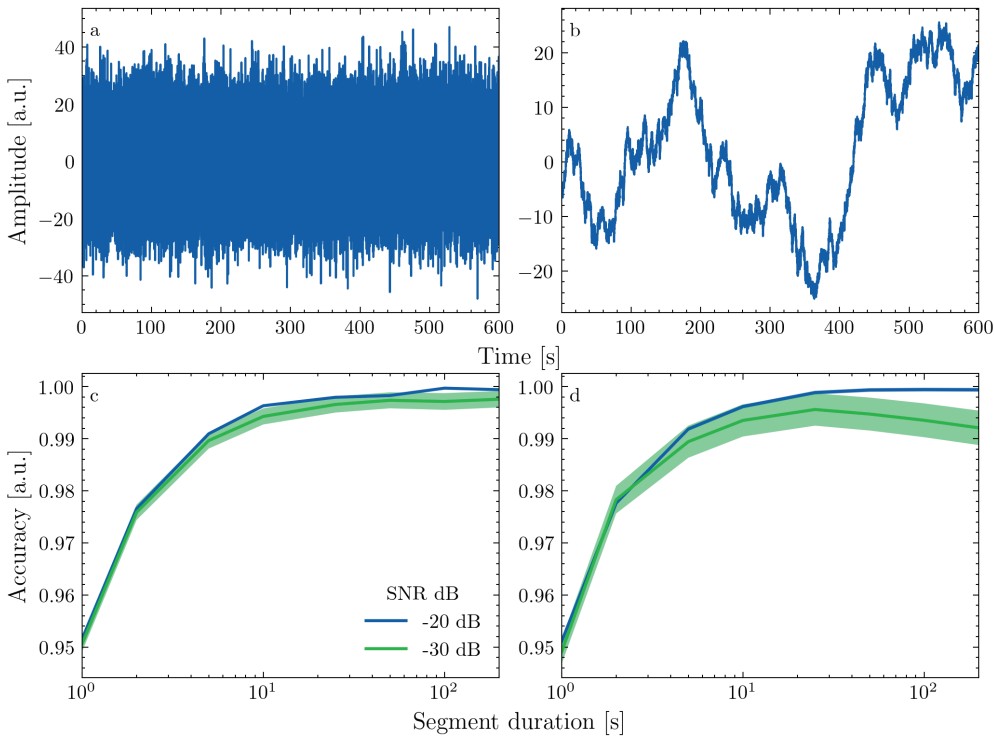

**Fig 2. Simulation results.**

## Results

The aim of using a generative model to simulate data was to examine the effect of segmentation on model accuracy against a known ground truth. The results show that, when the noise is Gaussian, segmentation into ever shorter bits gradually reduces model accuracy (Fig 2a). This makes perfect sense since the advantages of segmentation discussed earlier (representative sampling, robustness to outliers, ensuring stationarity), do not apply when dealing with homogeneous data from a stationary process. This trend is independent of the signal-to-noise ratio (SNR), although the model's variability increases at lower SNRs. However, when the noise is non-stationary, segmentation initially improves prediction accuracy if the SNR is sufficiently low (Fig 2b). Still, further segmentation decreases prediction accuracy, meaning that there is an optimal length where segments can be considered approximately stationary without substantially reducing model accuracy. Note that, since the simulation lacks any physiological plausibility, the absolute values contain no relevant information. For example, the absolute SNRs are very low because a lot of samples in the simulated response are 0, reducing the average power.

To test how segmentation affects models trained on actual neural data, we predicted EEG recordings of brain responses to continuous naturalistic speech from the spectrogram of the stimulus while dividing the data into ever shorter segments. To test how the effect of segmentation depends on the model's flexibility, we repeated the analysis for three different levels of spectral detail. Because prediction accuracy differed strongly between participants and models, we normalized each model's accuracy, for each participant, by the maximum for that participant, such that the results reflect the relative changes in model accuracy with segment duration. On average, decreasing segment duration from 120 s to 10 s improved

prediction accuracy by 10 to 15 percent (Fig 3a). The improvement was larger for the 16-band model compared to the 1- and 8-band model, suggesting that models with more more parameters may be more sensitive to non-stationary trends in the data. Decreasing segment duration below 5 s rapidly reduced prediction accuracy. The optimal value of the regularization parameter $\lambda$ was decreased with segment duration (Fig 3b). This shows that averaging across segments regularizes the model such that models fit on a larger number of short segments require less additional regularization via $\lambda$.

Because the effect of segmentation on model accuracy was highly variable across participants, we conducted a post-hoc analysis to investigate the source of this variability. We computed the difference in prediction accuracy between 10 s and 120 s segments for each participant to obtain an estimate of model improvement due to segmentation. We also took the highest prediction accuracy as an estimate of how well the participant's data could be predicted with the optimal combination of parameters. Linear regression revealed a negative relationship between these two estimates ($r = -0.62$, $p = 0.006$), suggesting that segmentation was especially effective for participants where model fitting was suboptimal (likely due to non-stationary trends and outliers in the data).

While segmentation improved model accuracy for most participants, it reduced accuracy by 10 percent for one of them. This may indicate that the recordings from the respective participants were unusually stationary. As our simulations showed, segmentation does not improve model but instead reduce model accuracy when the data are stationary.

As we elaborated earlier, one way in which segmentation can improve model accuracy is by reducing the effect of outliers - if the number of segments is increased, the effect of any single segment is reduced. To illustrate this point, we selected one of the participants where segmentation had a large effect and estimated the TRF for each segment individually after dividing the data into 120 and 5 (the optimal duration for that participant) second segments, respectively (Fig 4a, 4b). Here, we used only a single spectral band (i.e. the broadband envelope) to avoid arbitrarily selecting features. We identified outliers by sorting the segments by mean absolute TRF weight and selecting the largest five percent. We then computed the TRF for 120 and 5 second segments on the full data and with the outliers removed.

Fig 4c shows the respective TRFs for one fronto-central EEG channel with (solid line) and without (dashed line) outliers. While omitting 5% of outlier segments changed the mean absolute weight by 7% for 5 s segments, omitting the same amount of data changed the mean

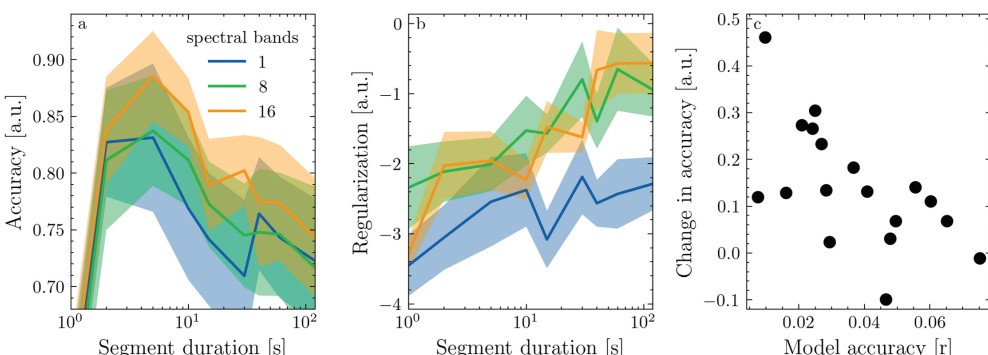

**Fig 3. Effect of data segmentation on models for EEG responses to speech.**

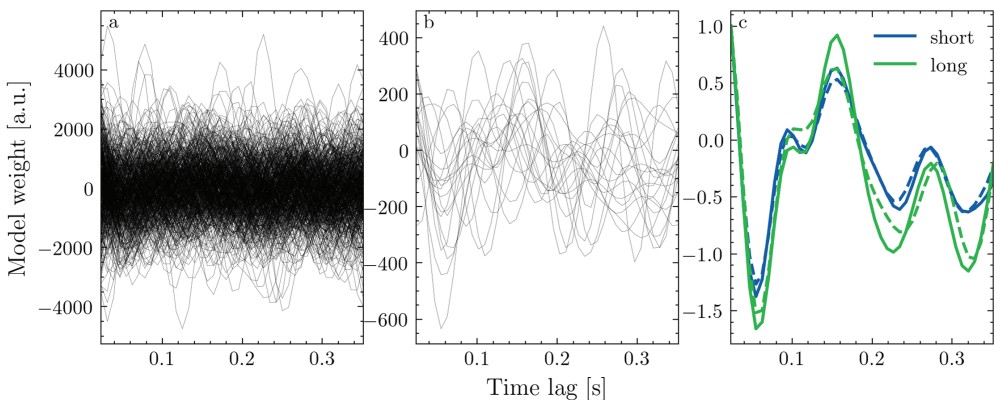

**Fig 4. Segmentation regularizes the effect of outliers.**

absolute weight by 13% for 120 s segments. This shows that increasing the number of segments can reduce the effect of outliers on the model's outcome. Note that this is merely an illustration, the optimal segment duration for isolating outliers will vary across recordings.

## Discussion

We investigated how speech encoding models are affected by non-stationary trends in neural recordings and how this depends on the length of the individual data segments. Our hypothesis was that, if the data where non-stationary, reshaping the data into smaller segments would improve model accuracy.

We used a generative simulation to test model accuracy under stationary and non-stationary noise. Under stationary noise, model accuracy decreased monotonically with segment duration. However, when the noise was non-stationary segmentation initially improved accuracy. To test our hypothesis on actual neural data, we used TRFs to predict EEG recordings of participants who listened to a narrated story. The results were qualitatively similar to the non-stationary simulation — segmentation improved model accuracy although very short segments were detrimental. The observed improvements were larger for the model that used a 16 band spectrogram, compared to the models using 8- and 1-band versions. Larger models could be affected more strongly because their flexibility could allow them to (over-)fit the non-stationary trends in the data.

Model improvements were observed mainly for segment durations between 2 and 20 seconds which is the same timescale where prior studies identified the transition of EEG recordings to non-stationarity [10,11,14]. Taken together, these findings indicate that long continuous neural recordings are indeed non-stationary and that segmentation improves model accuracy by making the data more stationary.

While we strongly suggest that future studies on speech encoding adopt data segmentation as part of their model fitting process, it should be noted that we analyzed data from a relatively small group of participants who all listened to the same stimulus material and were recorded with the same EEG setup. Thus, other studies should not blindly copy the parameters that yielded the best result in our study but instead test multiple segment durations.

Naturally, one wonders what may be the source of non-stationary trends in neural recording. A study that tested the non-stationarity of EEG recordings under different experimental

conditions found that there was no systematic difference between recordings during auditory stimulation, resting or cognitive tasks [15]. This suggests that non-stationarity is not directly related to perception or cognition but instead results from other, possibly non-physiological processes. This would also explain why we saw the largest improvements due to segmentation for participants where the baseline accuracy was low. However, this is speculative and would require additional studies where parameters of the recording setup are systematically varied.

Finally, one may object that dividing the data into segments of equal length is an overly simplistic way of dealing with deviations from stationarity in neural recordings. Maybe, there are extended segments where the recordings are perfectly stationary that would be unnecessarily interrupted by segmentation. There are more sophisticated algorithms that use wavelet transform, fractal dimensions or source separation to adaptively segment neural recordings based on their statistical properties [26–28]. However, implementation of these procedures requires substantial technical knowledge, while dividing the data into normalized segments of equal length can be done in two lines of code and has very little negative consequences.

## Conclusion

We demonstrated that segmentation improves the accuracy of TRF models that predict EEG recordings of brain responses to naturalistic speech. While long, continuous, recordings can provide a more naturalistic situation for speech comprehension, the resulting data should be reshaped into shorter segments prior to modeling. Ideally, one would choose the longest possible duration where the data can be considered (approximately) stationary. However, since this is difficult, the most practical solution may be to test multiple segment durations and choose the one that yields the highest accuracy.

## Author contributions

**Formal analysis:** Ole Bialas.

**Methodology:** Ole Bialas.

**Supervision:** Edmund C. Lalor.

**Visualization:** Ole Bialas.

**Writing – original draft:** Ole Bialas.

**Writing – review & editing:** Ole Bialas, Edmund C. Lalor.

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
