## [Decision Letter · Decision Letter 0]

8 Oct 2024

PONE-D-24-38627Appropriate data segmentation improves speech encoding modelsPLOS ONE

Dear Dr. Bialas,

Thank you for submitting your manuscript to PLOS ONE. After careful consideration, we feel that it has merit but does not fully meet PLOS ONE’s publication criteria as it currently stands. Therefore, we invite you to submit a revised version of the manuscript that addresses the points raised during the review process.

Editor Comments:Thanks for submitting your work to PLOS ONE. Your manuscript has now been assessed by our editorial team and external peer experts. While they found it interesting, you will see that they have raised many serious problems and are advising that you should revise your manuscript thoroughly. At the same time, please submit the point-by-point responses to reviewers' comments. If you are prepared to undertake the work required, I would be pleased to reconsider my decision. Please note that this revision decision does not assure the acceptance of your work. Thanks for the opportunity to consider your work. Editor Reviews:1.Please further re-evaluate and clarify the rationality of the methodology, which is linked to the result reliability and scientific rigor.2.Please consider to comprehensively propose the strengths and limitations of your work.3. Please consider to use structured abstract.4. I did not see conclusion section. Please add this part.5. Please consider to add some tables to summarize the core data of your analysis and results. I think it will be clearer for readers to grasp your findings.6. Minor suggestion: improve the quality of your figures (both in resolution and contents).==============================

We look forward to receiving your revised manuscript.

Kind regards,

Li Yang, M.D.

Academic Editor

PLOS ONE

Journal Requirements:

**Additional Editor Comments:**

Please address the concerns proposed by Editor Reviews.

**Reviewers' comments:**

Reviewer's Responses to Questions

**Comments to the Author**

1. Is the manuscript technically sound, and do the data support the conclusions?

Reviewer #1: Partly

Reviewer #2: Partly

2. Has the statistical analysis been performed appropriately and rigorously? 

Reviewer #1: Yes

Reviewer #2: I Don't Know

3. Have the authors made all data underlying the findings in their manuscript fully available?

Reviewer #1: Yes

Reviewer #2: No

4. Is the manuscript presented in an intelligible fashion and written in standard English?

Reviewer #1: Yes

Reviewer #2: Yes

5. Review Comments to the Author

Reviewer #1: Thank you for the opportunity to review this manuscript, which presents an important exploration of EEG responses to speech segmentation. I appreciate the authors’ efforts in conducting this research and believe that with some revisions, the paper has the potential to make a valuable contribution to the field. However, it is important that clear rationales for their methodological decisions are provided, as this will enhance the transparency and reproducibility of the findings. Detailed explanations of participant selection criteria, the choice of EEG data, and the specific segment lengths used would strengthen the overall clarity and impact of the manuscript. Additionally, the figures are almost publication-ready; they just need a few clarifying details.

I have attached a document with more detailed comments for your consideration. I look forward to reading future installments of this study.

Reviewer #2: In this paper the authors have used simulations and EEG recordings to investigate how non-stationarities affect temporal response functions (TRF) models for continuous speech processing. Based on the findings it is suggested that non-stationarities may reduce the performance of TRF models. However, it can be partially compensated by dividing the data into shorter segments that approximate stationary.

This is a topic of interest. For authors of this paper, I have the following comments.

1. The expression description of the paper needs improvement. The introduction section should be improved by clarifying the similarities and differences between the related work and the proposed method. In current form the contribution of the paper seems marginal. Authors should emphasize and clearly describe their contribution, preferably in the form of bullets.

2. Usually, a system used for evaluation purpose has a very careful design about the front-end sensors. However, no physical implementation and parameter detail is provided in this work.

3. The used materials and methods should be clearly described and clearly specify the implementation details/parameters of all methods.

4. Provide a URL of the studied dataset or share the studied dataset in csv format for an ease of review process.

5. Properly support claims in the discussion section by providing quantitative findings. Also present the performance comparison with counterparts, preferably in a tabular form while focusing on the main methods and major findings.

6. PLOS authors have the option to publish the peer review history of their article (what does this mean?). If published, this will include your full peer review and any attached files.

Reviewer #1: No

Reviewer #2: No

---

## [Author Response · Author response to Decision Letter 1]

12 Jan 2025

Dear Editor and Reviewers,

Thank you for providing us with the opportunity to refine and clarify our manuscript. In the following, we will address the points you have raised. Where it is appropriate, we will refer to edited lines in the revised manuscripts (edits are marked red).

Comments by the Editor

1.Please further re-evaluate and clarify the rationality of the methodology, which is linked to the result reliability and scientific rigor.

A: We focus on TRF models because they are the most commonly used kind of encoding model and they are comparatively small and simple (see second paragraph of the introduction). We analyze simulated data because they allow us to test our hypothesis under a known ground truth and we use EEG recordings to test our hypothesis on real data (see ”Methods” part of the abstract and the first paragraph in the ”Methods” section). We chose this data set because it has proven to be well suited for TRF analyses and the individual recordings are long enough to test our models across a wide range of segment durations (see section ”Modeling EEG responses to speech”).

2.Please consider to comprehensively propose the strengths and limitations of your work.

A: The strength of our approach lies in the combination of simulations and EEG analysis. This way we can test our hypothesis theoretically, under a known ground truth and empirically, on actual neural data. We added these considerations to the abstract as well as the discussion section (paragraph 4). Our results show that segmentation is an easy method for improving the accuracy of speech encoding models. This is relevant because these models are widely used in the neuroscience community, potentially including medical diagnostics.

3. Please consider to use structured abstract.

A: We changed the abstract to use a structured format.

4. I did not see conclusion section. Please add this part.

A: We have added a Conclusion section.

5. Please consider to add some tables to summarize the core data of your analysis and results. I think it will be clearer for readers to grasp your findings.

A: We could add table that contains the average model accuracies for each model and all segment durations. However, we think that the absolute model accuracy is not relevant here, since the key aspect is the relationship between model accuracy which is depicted in Fig. 3.

6. Minor suggestion: improve the quality of your figures (both in resolution and contents).

A: We re-created the figures with higher resolution.

Comments by Reviewer I

30-31: While read speech may approximate “real speech,” it still lacks many features of truly spontaneous speech. To substantiate this claim, consider citing the following studies: Howell, P., & Kadi-Hanifi (1991). Comparison of prosodic properties betweewn read and sponta- neous speech material. Speech Communication, 10(2), 163-169. https://doi.org/10.1016/0167- 6393(91)90039-V Mehta, G., & Cutler, A. (1988). Detection of target phonemes in sponta- neous and read speech. Language and Speech, 31( Pt 2), 135-156. https://doi.org/10.1177/0023830988031002

A: Many thanks to the reviewer for pointing us to this interesting literature. We fully agree that read speech may deviate from real conversational speech in important ways. However, in the case of our study, we are not making any argument about how speech is processed in the brain. Rather, we are investigating how the encoding models typically used in the speech literature are affected by the statistical properties of the neural recordings. Nonetheless, we have added a small caveat to the discussion on the nature of the speech materials we used.

47-49: Please clarify the definition of “estimand” to assist readers who may be unfamiliar with the term.

A: Rather than providing a definition, we added an example to clarify that predictor refers to speech features and estimand refers to the neural recording

121-122: Upon reviewing Di Liberto et al. (2015) and Broderick et al. (2018), neither article specifies the actual material read by the speaker. If the original text is unavailable, could details regarding the narrative style or genre be provided? Understanding the type of text used could impact the interpretation of results.

A: The story is ”The Old Man and the Sea”. We added this information to the subsection ”Modeling EEG responses to speech”.

What criteria were used for selecting the 19 participants in this study?

A: The subjects were all healthy neurotypical adults, we added this information to the subsection ”Modeling EEG responses to speech”. There were no other special criteria applied. The data were collected for Di LIberto et al., 2015 and Broderick et al., 2018 and are described in more detail in those studies.

Why was this particular EEG database chosen for the analysis?

A: We chose this data set because it has proven to be well suited for TRF analyses and the individual recordings are long enough to test our models across a wide range of segment durations.

131–136: The inclusion of the number of spectral bands as a variable is a valuable addition; however, they are infrequently referenced in the results and discussion sections. It would be beneficial to discuss their impact on the findings.

A: We found that the accuracy improvement from segmentation was larger for the 16-band model compared to the 8- and 1-band models. This could be because the increased number of parameters increases the extent to which the model (over-)fits the non-stationary trends in the data. We added this to our discussion section.

143-145: Please clarify the source of the “50 minutes” of data, as the analysis seems to involve only 19 subjects, each contributing between 170 and 200 seconds of data.

A: Thank you for pointing out the ambiguity here. There are 50 minutes of recordings for every subject, we rephrased the paragraph to make this clearer.

143-145: What was the rationale for choosing these specific segment lengths? Given that Gonen and Tcheslavski (2012) identified a critical duration of 2 to 25 seconds, why not test segments up to 25 seconds using the current dataset?

A: The durations were chosen so that the 50 minutes of data could be divided evenly into segments for every duration (we added this expla- 2 nation to the manuscript). The studies that identified critical durations between 2 and 25 seconds tested the randomness or gaussianity of the recording. Because our approach was different we chose to include the widest possible range of values to minimize a priori assumptions about the relevant timescale. However, our results seem to agree with the mentioned studies since there is little difference between using segments that are 20 seconds and longer. We added a sentence to both the Methods and Discussion sections to relate our segment durations to the previous studies

145-146: An exclusion criterion of 0.01 for model accuracy appears unusually low. Please provide justification for this threshold.

A: It is important to note that we are attempting to model unaveraged EEG - most of which is unrelated to the speech stimulus. It is quite typical for such models to produce prediction accuracy scores of r ¡ 0.1. The point of the exclusion criterion was to reject participants where the model could not explain any variance in the data. For the one subject that was rejected due to the criterion, accuracy approximated 0 for all models (average r was 0.001). We added this latter information to the paragraph

191-193: “On average” is stated twice in this sentence.

A: We corrected the sentence

198: It might be beyond the scope of this paper, but was any post hoc analysis conducted to determine the factors driving variability across subjects?

A: This was to motivation for the linear regression that relates the model improvement from segmentation to the baseline prediction accuracy. The fact that the improvement was larger for subjects where the baseline accuracy was low suggests that the non-stationarity is not related to stimulus driven activity. However, since interindividual variability in EEG recordings is another area of research, we do consider further investigations into this beyond the scope of this paper indeed.

202-204: The phrase, “What’s more, there was a negative relationship (linear regression, r = -0.62, p = 0.006) between the relative change in accuracy due to segmentation. . . ” needs clarification. Specifically, which segmentation is referenced here: the average of 10 and 120 seconds, the average across all segment durations, or the optimal segmentation for each sub- ject?

A: This is indeed ambiguously phrased. For every subject we took the relative difference in accuracy between 10 and 120 second segments (i.e. the improvement due to segmentation) and the peak prediction accuracy (i.e. how well the subjects EEG could be predicted with the best combination of parameters). These two estimates were negatively correlated across subjects. We rewrote the paragraph to make it clearer.

206–208: Is there any commentary regarding the identified outlier?

A: It’s possible that the recordings of the outlier were unusually stationary. As our simulations showed, segmentation does not improve but instead reduces model accuracy when the data are stationary. We added this commentary to the paragraph.

209: Consider using “improve” or “increase” instead of “affect” to more powerfully convey the direction of the change.

A: We changed the wording of the sentence.

Figures Although it is common in the field, including a definition for “a.u.” would be helpful for readers.

A: We added an explanation to the caption of each figure.

3 Please correct the x-axis label from “segmend” to “segment.”

A: We fixed the typo.

Figure 3a & c: Although both y-axes are represented in a.u., are they on the same scale? For instance, does a 0.1 change in accuracy in panel c correspond to a 0.1 change in accuracy in panel a? If so, it would be helpful to specify this in the figure legend.

A: No, they are not the same. Figure 3a shows accuracy normalized to each model’s maximum and 3c shows the relative difference between 10s and 120s segments. We modified the caption to make this more explicit.

Figure 3c: Regarding the “Change in accuracy,” which segmentation level was used to ob- tain these values?

A: This is the relative difference between 10s and 120s segments. Because this information would have been too long to put it in the axis title, we put it in the figure caption

While the Ethics statement is appreciated, it is somewhat confusing as neither author is affiliated with Trinity College Dublin. It would be helpful to include a sentence clarifying that the EEG data originated from Di Liberto et al. (2015) or Broderick et al. (2018) at that institution, allowing readers to understand the context without needing to consult those studies.

A: Personally, we think that an ethics section is not strictly necessary since we only used publicly available data. However, this seems to be a requirement by the journal. Your suggestion is a good way of making the relationship between our study and the data more explicit.

Comments by Reviewer2

1. The expression description of the paper needs improvement. The introduction section should be improved by clarifying the similarities and differences between the related work and the proposed method. In current form the contribution of the paper seems marginal. Authors should emphasize and clearly describe their contribution, preferably in the form of bullets.

A: There are two ways in which the paper contributes to the field: 1. We probe the non- stationarity of EEG recordings by quantifying the improvement of model accuracy due to data segmentation and by using simulations to show that these improvements can be explained by non- stationary trends in the data 2. We show that the common practice of fitting speech encoding models to long, continuous recordings is suboptimal and urge researchers to adopt segmentation as part of their model fitting process. Regarding the differences between ours and related work, see the last paragraph of the subsection ”Are neural recordings stationary?”. We also hope that the added structured abstract and conclusion sections add some clarity with respect of the points you raised.

2. Usually, a system used for evaluation purpose has a very careful design about the front-end sensors. However, no physical implementation and parameter detail is provided in this work.

A: The data we present in this manuscript came from a publicly available EEG dataset. We note that the data are described in two previously published papers (Di Liberto et al., 2015 Broderick et al., 2018). Those papers describe the data collection hardware in detail.

3. The used materials and methods should be clearly described and clearly specify the imple- mentation details/parameters of all methods.

A: We have attempted to be clear about the simulation and EEG analysis parameters throughout the methods and results sections. Also, as mentioned be- low, we have made all of our code available so that others can see precisely what parameters we used for each step. 4

4. Provide a URL of the studied dataset or share the studied dataset in csv format for an ease of review process.

A: The data is hosted on OpenNeuro and can be obtained, together with all of our code, from a GitHub repository. Both, repositories are linked in the ”Data availability” subsection in ”Materials and methods”. The data are shared in the standardized BIDS format — csv would be a very inefficient way of sharing a large EEG data set.

5. Properly support claims in the discussion section by providing quantitative findings. Also present the performance comparison with counterparts, preferably in a tabular form while focusing on the main methods and major findings.

A: We have made extensive revisions to the results and discussion sections and hope that both are now coherent in terms of the quantitative findings and the discussion of same.

Sincerely, Ole Bialas and Edmund Lalor

---

## [Decision Letter · Decision Letter 1]

3 Feb 2025

PONE-D-24-38627R1Appropriate data segmentation improves speech encoding modelsPLOS ONE

Dear Dr. Bialas,

Thank you for submitting your manuscript to PLOS ONE. After careful consideration, we feel that it has merit but does not fully meet PLOS ONE’s publication criteria as it currently stands. Therefore, we invite you to submit a revised version of the manuscript that addresses the points raised during the review process.

We look forward to receiving your revised manuscript.

Kind regards,

Li Yang, M.D.

Academic Editor

PLOS ONE

**Journal Requirements:**

Reviewers' comments:

Reviewer's Responses to Questions

**Comments to the Author**

1. If the authors have adequately addressed your comments raised in a previous round of review and you feel that this manuscript is now acceptable for publication, you may indicate that here to bypass the “Comments to the Author” section, enter your conflict of interest statement in the “Confidential to Editor” section, and submit your "Accept" recommendation.

Reviewer #1: All comments have been addressed

2. Is the manuscript technically sound, and do the data support the conclusions?

Reviewer #1: Yes

3. Has the statistical analysis been performed appropriately and rigorously? 

Reviewer #1: N/A

4. Have the authors made all data underlying the findings in their manuscript fully available?

Reviewer #1: Yes

5. Is the manuscript presented in an intelligible fashion and written in standard English?

Reviewer #1: Yes

6. Review Comments to the Author

**Reviewer #1:** Thank you for the thorough and thoughtful revisions. I appreciated the explicit outlining of how this study is distinct and contributes to the field, as well as the clear statement of the hypothesis. The explanation for the low data exclusion threshold and discussion of the outlier were also helpful. The concluding paragraph is a welcome addition and effectively summarized the study. Additionally, I appreciated the clarification regarding the incompatibility for comparing Figure 3 and a.u.s. Thank you for specifying that the reading passage was “The Old Man and the Sea.” For unfamiliar readers, could the authorship or a brief description of the story be added? This is particularly relevant as different emotional states may yield different EEG results. Good luck with this line of research!

7. PLOS authors have the option to publish the peer review history of their article (what does this mean?). If published, this will include your full peer review and any attached files.

Reviewer #1: No

---

## [Author Response · Author response to Decision Letter 2]

12 Mar 2025

Dear Reviewer 1,

Thank you for your comments. In your response, you requested:

"For unfamiliar readers, could the authorship or a brief description of the story be added?"

We have added two sentences to the section "Modeling EEG responses to speech" that describe the content and tone of the narration to allow the reader to assess the emotional impact.

Sincerely,

Ole Bialas and Edmund C. Lalor

---

## [Decision Letter · Decision Letter 2]

18 Mar 2025

PONE-D-24-38627R2Appropriate data segmentation improves speech encoding modelsPLOS ONE

Dear Dr. Bialas,

Thank you for submitting your manuscript to PLOS ONE. After careful consideration, we feel that it has merit but does not fully meet PLOS ONE’s publication criteria as it currently stands. Therefore, we invite you to submit a revised version of the manuscript that addresses the points raised during the review process.

We look forward to receiving your revised manuscript.

Kind regards,

Li Yang, M.D.

Academic Editor

PLOS ONE

**Journal Requirements:**

**Additional Editor Comments:**

Thanks for submitting your revised paper to PLOS ONE, and I am pleased to inform you that your paper has now been approved by the previous peer expert. But before I can recommend the final editorial decision to our journal office, some minor issues need your attention.

1) Revise your title and specify the detailed study type or brief summary on the study contents. For example: Appropriate data segmentation improves speech encoding models: A comprehensive analysis of ....., or anything you think suitable.

2) Please consider to use structured abstract including background, objective, methods, results, and conclusion, and each of them shoule be presented as a separate paragraph, instead of just putting them together.

3) I note that there is no any table in the paper. Please consider to add some tables to summarize your core data, and it is better for readers to follow your main findings.

Reviewers' comments:

Reviewer's Responses to Questions

**Comments to the Author**

1. If the authors have adequately addressed your comments raised in a previous round of review and you feel that this manuscript is now acceptable for publication, you may indicate that here to bypass the “Comments to the Author” section, enter your conflict of interest statement in the “Confidential to Editor” section, and submit your "Accept" recommendation.

Reviewer #1: All comments have been addressed

2. Is the manuscript technically sound, and do the data support the conclusions?

Reviewer #1: Yes

3. Has the statistical analysis been performed appropriately and rigorously? 

Reviewer #1: Yes

4. Have the authors made all data underlying the findings in their manuscript fully available?

Reviewer #1: Yes

5. Is the manuscript presented in an intelligible fashion and written in standard English?

Reviewer #1: Yes

6. Review Comments to the Author

**Reviewer #1: **Thank you for the opportunity to review the newly revised manuscript. I appreciate the description of the read speech, and my comments have all been successfully addressed. I have no further questions and wish the authors the best of luck with this project.

7. PLOS authors have the option to publish the peer review history of their article (what does this mean?). If published, this will include your full peer review and any attached files.

Reviewer #1: No

---

## [Author Response · Author response to Decision Letter 3]

25 Mar 2025

Dear Dr. Yang,

Thank you for your comments. We have added a summary to the title, it now reads:

Appropriate data segmentation improves speech encoding models: Analysis and simulation of

electrophysiological recordings. We also replaced ”Introduction” with ”Background” in the structured abstract

and put each point on a separate paragraph. In your last point, you requested that we

add a table to our publication to make it easier for the reader to follow the main findings.

We believe that adding a table that compares the accuracy of the different encoding models with

the different segment durations would likely be confusing and detracting rather than beneficial to

readers. That is because the key point is the qualitative change in model accuracy that results from

a change in segment duration, not the absolute values. We believe that the figures, combined with

the text, illustrate this point well.

Sincerely,

Ole Bialas and Edmund Lalor

---

## [Editor Report · Decision Letter 3]

26 Mar 2025

PONE-D-24-38627R3Appropriate Data Segmentation Improves Speech Encoding Models: Analysis and Simulation of Electrophysiological RecordingsPLOS ONE

Dear Dr. Bialas,

Thank you for submitting your manuscript to PLOS ONE. After careful consideration, we feel that it has merit but does not fully meet PLOS ONE’s publication criteria as it currently stands. Therefore, we invite you to submit a revised version of the manuscript that addresses the points raised during the review process.

We look forward to receiving your revised manuscript.

Kind regards,

Li Yang, M.D.

Academic Editor

PLOS ONE

Journal Requirements:

**Additional Editor Comments:**

Thanks for your response to my concerns. However, the abstract still can not meet the standard for publication. First, each section should be presented as a separate paragraph instead of just putting them together. Second, you should simplify the abstract since there are too many words included, such as background, methods, conclusion. Please just present the core points. Third, you should annotate the abbreviations in the abtract, such as 

**Your current abstract in the manuscript:**

**Background:** In recent decades, studies modeling the neural processing of

continuous, naturalistic, speech provided new insights into how speech and

language are represented in the brain. However, the linear encoder models

commonly used in such studies assume that the underlying data are

stationary, varying to a fixed degree around a constant mean.

    Long, continuous, neural recordings may violate this assumption leading

to impaired model performance.** Objective:** Our objective was to examine

the effect of non-stationary trends in continuous neural recordings on the

performance of linear speech encoding models.

**    Methods:** We used temporal response functions (TRFs), to predict

continuous neural responses to speech while splitting the data into segments

of varying length, prior to model fitting. Our Hypothesis was that, if the

data were non-stationary, segmentation should improve model performance

by making each segment approximately stationary.

    We used generative models for simulate and predict stationary and

non-stationary neural data, testing our hypothesis under a known ground

truth. We then used encoding models to predict the EEG recordings of

participants who listened to a narrated story, testing our hypothesis on

actual neural data. **Results:** Simulations showed that, for stationary data,

increasing segmentation steadily decreased model performance. For

non-stationary data however, segmentation initially improved model

performance. Modeling of EEG recordings yielded similar results: segments

of intermediate length (5-15 s) led to improved model performance

compared to very short (1-2 s) and very long (30-120 s) segments.

    **Conclusions:** We showed that data segmentation improves the

performance of encoding models in predicting both simulated and real

neural data and that this can be explained by the fact that shorter segments

approximate stationarity more closely. We thus conclude that the common

practice of applying encoding models to long continuous segments of data is

suboptimal and implore researchers to segment their data prior to model

fitting.

**My recommend abstract format:**

**Background: **In recent decades, studies modeling the neural processing of continuous, naturalistic, speech provided new insights into how speech and language are represented in the brain. However, the linear encoder models commonly used in such studies assume that the underlying data are stationary, varying to a fixed degree around a constant mean. Long, continuous, neural recordings may violate this assumption leading to impaired model performance.

**Objective:** We aimed to examine the effect of non-stationary trends in continuous neural recordings on the performance of linear speech encoding models. 

(This part can be merged into the introduction as well. Remember to delete the 'Objective:' if you choose to merge.)

**Methods: **We used temporal response functions (TRFs), to predict continuous neural responses to speech while splitting the data into segments of varying length, prior to model fitting. Our Hypothesis was that, if the data were non-stationary, segmentation should improve model performance by making each segment approximately stationary. We used generative models for simulate and predict stationary and non-stationary neural data, testing our hypothesis under a known ground truth. We then used encoding models to predict the EEG recordings of participants who listened to a narrated story, testing our hypothesis on actual neural data.

**Results:** Simulations showed that, for stationary data, increasing segmentation steadily decreased model performance. For non-stationary data however, segmentation initially improved model performance. Modeling of EEG recordings yielded similar results: segments of intermediate length (5-15 s) led to improved model performance compared to very short (1-2 s) and very long (30-120 s) segments.

**Conclusions:** Our analysis showed that data segmentation improves the performance of encoding models in predicting both simulated and real neural data and that this can be explained by the fact that shorter segments approximate stationarity more closely. Thus, the common practice of applying encoding models to long continuous segments of data is suboptimal and implore researchers to segment their data prior to model fitting.

---

## [Author Response · Author response to Decision Letter 4]

1 Apr 2025

Dear Dr. Yang,

Thank you for providing an example of the desired formatting. We had inserted new paragraphs in the revised abstract, but the PLOS Latex template does not include a blank line after a paragraph. We changed the formatting of the abstract by manually inserting a blank line after every point. We also followed your suggestion to merge the ”Objectives" into the ”Background” section. We also made some minor changes for conciseness and hope that this meets the standards for publication.

Sincerely,

Ole Bialas and Edmund Lalor

---

## [Editor Report · Decision Letter 4]

6 Apr 2025

Appropriate Data Segmentation Improves Speech Encoding Models: Analysis and Simulation of Electrophysiological Recordings

PONE-D-24-38627R4

Dear Dr. Bialas,

We’re pleased to inform you that your manuscript has been judged scientifically suitable for publication and will be formally accepted for publication once it meets all outstanding technical requirements.

Kind regards,

Li Yang, M.D.

Academic Editor

PLOS ONE

Additional Editor Comments (optional):

Thanks for the authors' efforts to comprehensively improve your manuscript according to editor's and reviewers' comments. I am pleased to inform you that your paper can be accepted for publication now. Thanks for the chance to assess your interesting and important work. Additionally, many thanks for all the reviewers' precious inputs.
---

## [Editor Report · Acceptance letter]

PONE-D-24-38627R4

PLOS ONE

Dear Dr. Bialas,

I'm pleased to inform you that your manuscript has been deemed suitable for publication in PLOS ONE. Congratulations! Your manuscript is now being handed over to our production team.

Kind regards,

on behalf of

Dr. Li Yang

Academic Editor

PLOS ONE